

# Impact of community-based forest restoration on stand structural attributes, aboveground biomass and carbon stock compared to state-managed forests in tropical ecosystems of Sri Lanka

Shahzad Ahmad[1], Haiping Xu[1] and E. M. B. P. Ekanayake[2]

[1] International Business School of Hainan University, Hainan University, Haikou, China
[2] Department of Forest Conservation, Sampathpaya, Battaramulla, Sri Lanka

## ABSTRACT

Estimation of plant community composition, aboveground biomass and carbon stock is crucial for understanding forest ecology, strengthening environmental management, and developing effective tools and policies for forest restoration. This study was conducted in nine different forest reserves in Sri Lanka from 2012 to 2018 to examine the impact of community-based forest restoration (CBFR) on stand structural attributes, aboveground biomass, and carbon stock compared to state-managed forests. In total, 180 plots (90 plots in community-managed restoration blocks (CMRBs) and 90 plots assigned to state-managed restoration blocks (SMRBs)) were sampled at the study site. To conduct an inventory of standing trees, circular plots with a radius of 12.6 m (equivalent to an area of 500 square meters) were established. The Shannon diversity index, Allometric equations and Difference in Differences (DID) estimation were used to assess the data. Our study provides evidence of the positive impact of the CBFR program on enriching trees diversity. Considering stand structural attributes of both blocks showed higher trees density in the smaller diameter at breast height (DBH) category, indicating growth in both CMRBs and SMRBs. The results showed that tree biomass and carbon density were disproportionally distributed across the nine different forest reserves. On average, tree biomass and carbon density were higher in SMRBs (79.97 Mg ha$^{-1}$, 37.58 Mg C ha$^{-1}$) compared to CMRBs (33.51 Mg ha$^{-1}$, 15.74 Mg C ha$^{-1}$). However, CMRBs in Madigala reserve represent the highest biomass (56.53 and 59.92 Mg ha$^{-1}$) and carbon density (26.57 and 28.16 Mg C ha$^{-1}$). The results of biomass and carbon estimates were higher in all SMRBs in the nine different forest reserves compared to CMRBs. The findings suggest that future forest restoration programs in Sri Lanka should enhance participatory approaches to optimize tree species diversity, density and carbon storage, particularly in community-controlled forests. Our findings could assist developing tropical nations in understanding how CBFR impacts forest restoration objectives and improves the provision of ecological services within forests.

Corresponding author
Haiping Xu,
xuhaiping@hainanu.edu.cn

# INTRODUCTION

Globally, forests are identified as one of the crucial ecological components for maintaining and supporting the social-ecological systems, particularly in rural areas, where forestry serves as the primary land-use type in areas, playing a crucial role in managing the rural socio-ecological system and shaping the landscape (*Sunderland et al., 2017*). One billion local people are responsible for managing 15.5% of the world's forests, and this number is rising (*Rights and Resources Initiative, 2014*). Specifically, tropical forests provide livelihoods for approximately 410 million people, including 60 million indigenous people dependent on forest resources (*World Bank, 2004*). Thus, the management of forestland without consideration of public opinion, particularly on land use and forest policies, is largely ineffective and suffers from insufficient scientific direction and weak enforcement (*Rametsteiner & Whiteman, 2014*). In addition, studies in countries like Ecuador, Philippines, Zambia and Indonesia found that the existing "command and control" approach of forest protection, or managed by the state without considering local people has not been effective in ensuring the sustainability of natural forests (*Ningsih, Ingram & Savilaakso, 2020*; *Fischer et al., 2023*).

Moreover, the fact that so many people depend on forest areas suggests that there is an intrinsic relationship between forests and local people. A study conducted by *Persha, Agrawal & Chhatre (2011)* in six countries across East Africa and South Asia found that the engagement of local people in forest-related decision-making and management could enhance forest conditions because they possess indigenous knowledge regarding the local environment, which can be used to develop and implement effective management strategies. Furthermore, their easy accessibility to the forest provides them with a relatively great advantage for monitoring forest resource use. In addition to this, in several cultures, forests are consciously protected as sacred places due to their traditional, cultural, and spiritual beliefs (*Ormsby & Bhagwat, 2010*; *Ormsby, 2012*; *Lowman & Sinu, 2017*). Moreover, local communities have an invested interest in conserving and utilizing forests to meet community needs (*Brown, Gillespie & Lugo, 1989*; *Béné et al., 2009*). Therefore, community forestry is particularly widespread in Asia and Latin America, where communities and indigenous people manage a substantial portion of forest land, including 25% of the forest area in the Asia-Pacific region (*Sunderlin, Hatcher & Liddle, 2008*).

The original idea behind Community-Based Forest Restoration (CBFR) was to improve forest conditions and support poor rural communities in the sustainable utilization of forest resources in their vicinity as a livelihood asset (*Gurung et al., 2013*). Several studies have highlighted the impact of CBFR on livelihoods as well as forest ecosystems. A study conducted by *René Oyono (2005)*, *Pulhin & Dressler (2009)*, *Beauchamp & Ingram (2011)* highlighted the impact of community-based forest activities on rural livelihoods in terms of empowering women, reducing rural poverty, improving livelihoods, and promoting

sustainable forest management. On the other hand, impact on increasing carbon sequestration (*Luintel, Bluffstone & Scheller, 2018*), reducing emissions (*Pelletier, Gélinas & Skutsch, 2016*), protecting and improving the forest ecosystem (*Acharya, 2003*; *Gurung et al., 2013*; *Bijaya et al., 2016*; *Chinangwa, Pullin & Hockley, 2017*; *Chowdhury et al., 2018*; *Ekanayake, Cirella & Xie, 2020*) also mentioned in the literature. In this context, CBFR programs have become an established component of international forest policy. Conversely, state-controlled forests impose control and restriction policies that mostly negatively affect forest dwellers' economies and livelihoods. In some cases, these policies also lead to the overexploitation of forests and their resources, drawing the conclusion that sometimes state-managed forest management approaches are ineffective (*Buffum, 2012*; *Agrawal et al., 2013*; *Ekanayake et al., 2018*). For instance, a meta-study conducted by *Hayes (2006)* examined data of 13 countries, comparing the relationship between vegetation density and the presence of rules in 76 state-managed park areas and 87 community-managed non-park areas found that in the community-managed non-park areas, 60% had identified rules for all forest products which is comparatively higher than in the state-managed park areas (30%). Also, this study found that in 30% of the state-managed parks, floral density was abundant, with an average of 36% and sparse of 44%. At the same time, in the community-managed areas, tree cover was abundant at 29%, with an average of 43% and sparse cover of 28%. A similar result was found in another meta-study conducted by *Porter-Bolland et al. (2012)* in the state protected area and community-managed areas. They observed that in the publicly protected area, the mean annual forest cover change was −1.47% and it was −0.24% in the community-managed area.

In comparison to other developing countries, the expansion of population size, political instability, and harsh economic conditions have resulted in endless pressure on forests and forest resources in Sri Lanka (*Palo & Mery, 1996*; *Niesenbaum, Salazar & Diop, 2005*). National figures show that between 1956 and 2010, the overall amount of forest cover decreased by 14.5%, and in 2010, 29.7% of the Sri Lankan land was covered by forest (*Fernando, 2017*). Hence, in line with the concept of community forestry, Sri Lanka has greater potential for implementing and expanding a worthy CBFR program. Community-managed forests, together with community-established plantations and woodlots, are among the most important components of tropical ecosystems in Sri Lanka, in addition to natural forests and state-managed forest plantations (*Béné et al., 2009*). The Sri Lankan government has modified and revised its forest management policies since the late 1980s. So far, Sri Lanka has implemented CBFR programs, with expanding project sites, an enlarged rural population involved, and increasing research. Currently, 167 community-based forest restoration sites have been established in Sri Lanka, and approximately 23,500 hectares of forest land are managed by the community. Studies have highlighted that the majority of Sri Lanka CBFR sites are located in semi-mixed evergreen forests (*Ekanayake, Cirella & Xie, 2020*; *Ekanayake et al., 2022*). Semi-mixed evergreen forests are found in the dry and intermediate zones of Sri Lanka. These forests are home to the richest biodiversity and contribute substantially to the global carbon budget (*Chheng et al., 2016*). Unfortunately, semi-mixed evergreen forests are affected more by shifting

cultivation, human disturbance, and illegal logging (*FAO, 2010a*; *Tripathi & Tripathi, 2010*; *Robinson, Holland & Naughton-Treves, 2014*). Therefore, the government established CBFR in semi-mixed evergreen forests to reduce anthropogenic pressure and enhance the livelihoods of forest-dependent people (*Fernando, 2017*). Several historical studies (*Bandaratillake, Durst & Bishop, 1995*; *Gunatileke & Chakravorty, 2003*; *Farooq et al., 2021*; *Ekanayake et al., 2022*) have shown the impact of community-based forest activities on rural livelihoods and forest ecosystems. A study conducted by *Dissanayake (1998)*, *Zoysa & Inoue (2016)*, *Ekanayake et al. (2022)* in Sri Lanka highlighted that CBFR increases household income. Moreover, *Ekanayake et al. (2022)* also revealed that rural residents who participate into CBFR programs have more saving, more informal education opportunism, and access to state land holdings than other residents who do not participate in CBFR programs. Previous studies have also highlighted that the impact may vary with the pre-existing conditions of rural society and forest ecosystems (*Ekanayake, Cirella & Xie, 2020*).

Consistent with previous studies of community-based forestry in Sri Lanka, the majority of them focused on the impact assessment of livelihood rather than on forest ecosystems. The most recent study (*Ekanayake, Cirella & Xie, 2020*) on the impacts of community forestry on forest conditions highlighted the impact in terms of species diversity and evidence of the human disturbances. They found that the count of invasive species is considerably less in community managed forest blocks than state managed blocks. However, no study conducted on with the special emphasis on woody species composition, structure and carbon stocks. Woody plants are perennials that produce secondary growth in the form of wood (*Zimdahl, 2018*). Moreover, it is defined as a perennial plants (trees, shrubs and lianas) characterized by the presence during the non-growing season of persistent aboveground dormant parts, as well as ligneous material (mainly in vascular tissues) (*Bettenfeld et al., 2020*). Therefore, this study on the impact assessment of CBFR on forest tree species diversity, density, biomass, and carbon stock is novel to Sri Lanka community-based research field. Despite accounting for a large proportion of the total vegetation (approximately four-fifths of the country's total vegetation), the semi-mixed evergreen forest type has received less research attention compared to other forest types (*Dittus, 1977*; *Forest Department, 2016*). Some studies try to address the species diversity in small forest patches with semi-evergreen vegetation. However, there is no large-scale study covering the impact of CBFR on woody species composition, structure, and carbon stock of semi-mixed evergreen forests. This forest that are found not only in Sri Lanka, but also throughout the tropical and subtropical regions of the world. However, there is limited study related to the impact of community-based restoration activities on global semi-mixed evergreen forest (*Chheng et al., 2016*). Hence, this study provides data to fill the gap in knowledge related to the semi-mixed evergreen forest in Sri Lanka and globally (*Mattei Faggin, Hendrik Behagel & Arts, 2017*). Therefore, the present study aims to assess the impact of community-based forest restoration on forest tree species diversity, density, biomass, and carbon stock in semi-mixed evergreen forests in Sri Lanka. We hypothesize that community-managed forest sites will have higher tree species diversity, density, biomass, and carbon stocks compared to state-managed sites. To accomplish these goals,

the following research questions were formulated: (1) How does the woody species diversity and density of selected semi-mixed evergreen forests compare? (2) What are the calculated tree biomass and carbon stocks in these forests? (3) How do these parameters differ between community-based forest sites and state-managed sites in Sri Lanka?

# MATERIALS AND METHODS

## Study area

Sri Lanka is a tropical island with a land extent of approximately 65,610 square kilometers. The country is divided into three climatic zones based on seasonal rainfall, natural resources, and agricultural land use: dry, wet, and intermediate zones (*World Bank, 2021*). The intermediate zone is a climatic zone and that lies between the wet and dry zones, was selected for the study (Fig. 1). The intermediate zone extends to nine out of the 25 administrative districts in Sri Lanka. The area covered by this zone is estimated to be around 1.2 million hectares, which is 13.2% of the country. Four administrative districts were purposefully selected from the intermediate zone, namely Badulla, Monaragala, Kandy, and Kurunegala, and nine CBFR sites located in these districts were further purposefully selected. In the intermediate zone, there is a relatively short and mild dry season, with an average annual rainfall ranging from 1,750 to 2,500 mm and an annual average temperature of 30 °C. The dominant soil types in the area were reddish-brown latosols and red-yellow podzolic soils, along with immature brown loams. The dominant vegetation type in the intermediate zone is semi-mixed evergreen forest, covering approximately 221,977 hectares (ha) of the land area. Evergreen species (*e.g.*, *Manilkara hexandra*, *Diospyros ebenum*, *Syzygium* spp.) dominate the forests within the study area, whereas the proportion of deciduous species (*e.g.*, *Chloroxylon swietenia* and *Vitex altissima*) is comparatively lower. Specifically, the southeastern and northwestern regions of the central Sri Lanka exhibited a higher prevalence of deciduous trees compared to the central and northern areas (*Gunatilleke & Gunatilleke, 1984*). The semi-mixed evergreen forest ecosystem of Sri Lanka harbors a diverse array of wildlife species, including 25 endemic bird species and 13 endemic mammal species (*Wikramanayake et al., 2022*). These forests occupy 20–30 plant families, 40–50 genera, and 40–60 species, with 17% of them endemic. *Euphorbiaceae, Moraceae, Anacardiaceae* and *Sapindaceae* are among the dominant vegetation composition observed in these forests (*Illangasinghe, Fujiwara & Saito, 1999*). The semi-mixed evergreen forest covers the approximately four-fifths of country vegetation, and these forests provide a wide range of provisioning services in the shape of non-timber forest products *i.e.*, medicinal products, yams, bee honey, fruits, nuts, rattan, bamboo, flowers, leafy vegetable, fodder and agriculture products (roping materials, stalks, green manure). According to *Ekanayake, Cirella & Xie (2020)*, vegetable cultivation, paddy farming, and shifting (Chena) cultivation constitute the primary farming activities in the intermediate zone, while 65–75% of households in the area rely on forests for their everyday necessities (*Liyanaarachchi, 2004*).

The main reason these four districts and nine CBFR sites were chosen because they are located in the same eco-climatic zone and represent similar major woody species compositions, namely semi-mixed evergreen forests. Furthermore, these natural forests

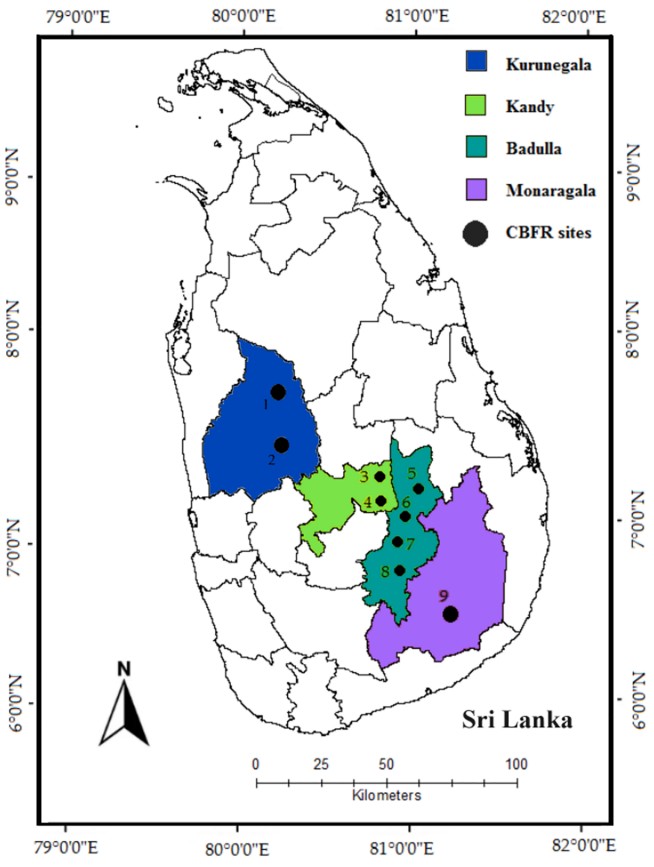

Study area in Semi-mixed evergreen forest in Sri Lanka.

**Figure 1  Location of the study area in a semi-mixed evergreen forest in Sri Lanka.** Community-based forest restoration (CBFR) sites in Sri Lanka (*Ekanayake et al., 2022*).

face various threats such as fires, grazing, illegal logging, and agricultural activities. Table 1 shows the forest location and forest cover of each CBFR site. All these nine CBFR sites, were established in nine natural forests (Table 1). These natural forests are divided into two forest blocks namely SMRBs and CMRBs. In each forest, the land area which is in and near the boundary of the forest and degraded due to anthropogenic pressure are managed as CMRBs, and forest land in the core of each forest area is managed as SMRBs.

## Data collection

The study followed a semi-experimental before-after control-impact (BACI) design (*Greene, 2003*). According to *Bowler et al. (2012)*, the BACI design is the most effective study design for assessing the impact of forest community management programs. Thus, this study used BACI design to estimate the impact on woody species composition and structure, aboveground woody biomass (AGWB), and carbon stock (CS). In this framework, the forest areas currently under CMRBs were taken as treatment group while the areas managed by SMRBs are regarded as the control group Furthermore, data

**Table 1 Districts, forest names and total area of nine CBFR sites.**

| Districts | Coordinates | Natural forest name | CBFR site names | Total area of the forest (hectares) |
|---|---|---|---|---|
| Kurunegala | 7°45′N 80°15′E | Dolukanda natural forest | Seeradunna | 7,713 |
| Kurunegala | | Rakaula natural forest and plantation | Aludeniyaya | 900 |
| Badulla | 6°59′05″N 81° 03′23″E | Madigala natural forest | Kinniyarawa | 300 |
| Badulla | | Dunukewala natural forest | Dunukewala | 237 |
| Badulla | | Walasgala aluyatawala natural forest | Walasgala | 70 |
| Badulla | | Gedaboyaya natural forest | Gedaboyaya | 50 |
| Kandy | 7°15′N 80°45′E | Galkanda natural forest | Wegala | 60 |
| Kandy | | Bambarabedda Waliketiya Mukalana Forest | Bambarabedda | 69 |
| Monaragala | 6°40′N 8°20′E | Hawanarawa natural forest | Hawanarawa | 50 |

collected in 2012 was classified as a before-CBFR program, while data gathered in 2018 was categorized as an after-CBFR program.

The total number of sample plots per tree stand depended on the stand's uniformity and size. At the research site, we sampled 180 individual plots in accordance with the operational guidelines for community forest management based on *Vianna & Fearnside (2014)*. These plots were distributed, with 90 plots allocated to the CMRBs and the remaining 90 plots assigned to the SMRBs. To conduct an inventory of standing trees with a diameter at breast height (DBH) greater than 5 cm, circular plots with a radius of 12.6 m (equivalent to 500 square meters) were established as the primary sites (Fig. 2). In each plot, a comprehensive inventory was conducted to count all trees, woody shrubs, and lianas. Subsequently, woody trees with a DBH (1.3 m) greater than 5 cm were measured using a DBH tape and recorded. For carbon stock measurements, Global Forest Resources Assessment guidelines were used during data collection in the field (*FAO, 2010b*).

During the study, forest field assistants and local people from the CBFR sites who possessed a deep understanding of the local flora played a vital role in identifying and documenting the common names of all plant species. During the study, data from 2012 was sourced from the documented inventory data of the Forest Department (FD). To collect 2018 data, we revisited the identical sample plots by utilizing the recorded GPS coordinates, ensuring consistency and comparability with the previous data collection.

## Data analysis
STATA version 15.1 was used for descriptive and inferential statistical analysis of data. The data analysis involved the use of simple descriptive statistics, including measures such as averages and percentages, as well as econometric analysis to examine and evaluate the data. Simple descriptive statistics was used to present the data of woody species density and structure (DBH class distribution). Woody species diversity, aboveground woody biomass, and carbon stock were determined using the following statistical methods: (a) The Shannon diversity index; (b) allometric equations; and (c) difference-in-differences (DID).

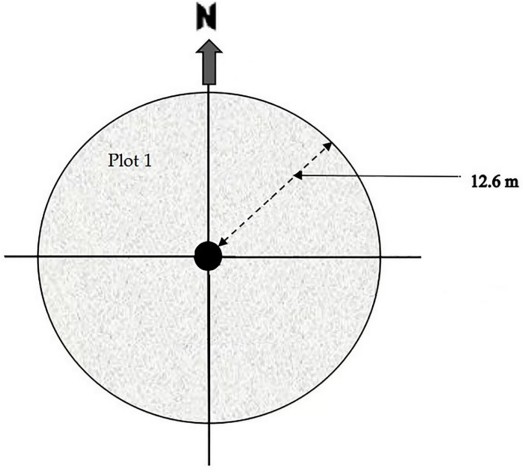

Forest plot layout sampling

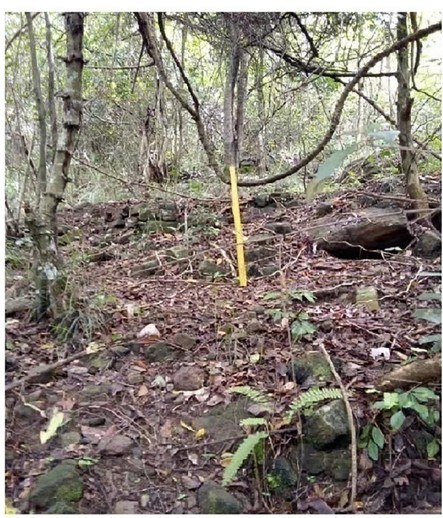

A center pole in sample plot

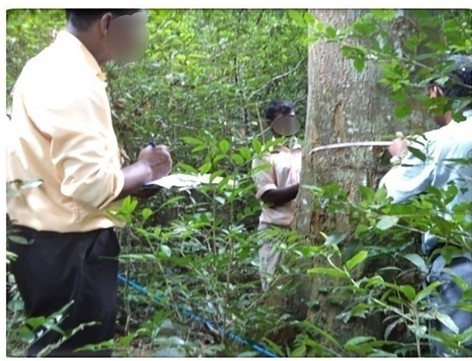

State-managed restoration blocks
(SMRBs)

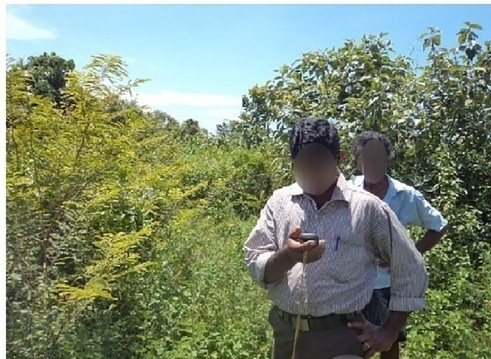

Community-managed restoration blocks
(CMRBs)

Woody species diversity, density, biomass and carbon stocks assessment

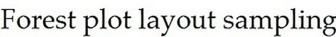

**Figure 2 Forest plot layout sampling and data collection for woody species diversity, density, biomass and carbon stocks assessment.**

### Woody species diversity analysis

Plant community variables such as tree stand density, basal area, and frequency were calculated. The Shannon diversity index was used to calculate the species diversity of woody plants (Eq. (1)). These indices are commonly used in analyzing the woody species diversity in community-based forest management systems (*Dhakal, Kafle & Yadava, 2011*; *Pandey et al., 2014*).

$$\text{Shannon diversity index (H)} = \sum_{i=1}^{s} -(\text{Pi} * \ln\text{Pi}) \tag{1}$$

where "$P_i$" represents the proportion of ith species in the entire population; "S" denotes the total number of species recorded in the sites; $\Sigma$ is the sum from species 1 to species "S";"*ln*" is the natural logarithm.

### Aboveground woody tree biomass and carbon stock analysis

Following the Global Forest Resources Assessment guidelines (*FAO, 2010b*), aboveground biomass (AGB) is defined as the total living biomass above the soil, encompassing seeds, foliage, bark, branches, stumps and stems. The study calculated aboveground woody tree biomass. We estimated the aboveground biomass of individual trees using an allometric model published by *Brown, Gillespie & Lugo (1989)* (Eq. (2)) and then summed it to obtain the final estimate (Eq. (3)).

$$\widehat{Y} = 34.4703 - 8.0671 \, (D) + 0.6586 \, (D^2) \tag{2}$$

where, $\widehat{Y}$ is the aboveground woody tree biomass (Individual tree biomass) (Kilogram Dry Matter/tree) and D is the diameter (cm). After calculating the aboveground biomass of individual trees, the total aboveground biomass is calculated in megagrams per hectare (Mg ha$^{-1}$), and the estimation is done on a per-hectare basis, as shown below (Eq. (3)). In this equation, we calculated the aboveground biomass (AGB) by summing the product of the unit biomass (AU) divided by 1,000 and multiplying it by the factor 10,000 divided by the plot area.

$$AGB = \left( \sum AU/1{,}000 \right) * (10{,}000/Plot area) \tag{3}$$

where AGB represents the aboveground tree biomass, measured in megagrams of dry matter per hectare (Mg DM ha$^{-1}$); AU represents the total tree biomass of all trees within the plot, measured in kilograms of dry matter per unit plot area (kg DM/plot area); Factor 1,000 equals the conversion of sample units from kilograms of dry matter per mega grams (kg DM/Mg);

The DM Factor 10,000 equals to conversion of the area from square meters (m$^2$) to hectares.

The Forest Resources Assessment guideline recommended a carbon conversion factor of 0.47 to estimate the carbon stock in aboveground biomass (*FAO, 2010b*). Following; *Jew et al. (2016)*, *Salas Macias, Alegre Orihuela & Iglesias Abad (2017)*, The carbon stock in aboveground biomass was calculated using the following equation (Eq. (4)).

$$\Delta AGB = (AGB * CF) \tag{4}$$

where $\Delta AGB$ represents the carbon content in the aboveground biomass measured in megagrams of carbon per hectare (Mg C ha$^{-1}$).

AGB refers to the aboveground tree biomass, measured in megagrams of dry matter per hectare (Mg DM ha$^{-1}$).

CF represents the carbon fraction expressed in megagrams of carbon per metric ton of dry matter (Mg C/t DM).

The default value is 0.47.

### Difference in differences analysis

The DID approach was used to assess the impact of the CBFR program on woody species diversity, density, biomass, and carbon stock. The DID approach is an analytical method that assesses the differential effects of an intervention over time by comparing the outcomes between the treatment group and the control group (*World Bank, 2023*). The DID model estimates causal effects in non-experimental settings that involve two-time periods, where a group of treated units receives a treatment of interest starting in the second period, whereas a comparison group remains untreated in both periods (*Roth et al., 2023*). In this study, the DID analysis helps us answer the counterfactual question: To what extent does the CBFR policy intervention impact the biomass and carbon stock of the community-managed restoration blocks of forest and state-managed restoration blocks of semi-mixed evergreen forest over a period of time? Therefore, the DID-based model used in this study can be written as Eq. (5).

$$Y = \alpha_0 + \alpha_1 \text{CBFR}^{\text{post}} + \alpha_2 \text{CBFR}^{\text{Tr}} + \alpha_3 \text{CBFR}^{\text{post}}\text{CBFR}^{\text{Tr}} + \varepsilon \qquad (5)$$

where the Y refers to the aboveground woody biomass/carbon stock, $\alpha$ is the DID coefficient estimate, $\alpha_1$ of $\text{CBFR}^{\text{post}}$ is the core coefficient equal to the time dummy (*i.e.*, after introducing the program), $\alpha_2$ of $\text{CBFR}^{\text{Tr}}$ represents the treatment group (*i.e.*, treatment sites), $\alpha_3$ of $\text{CBFR}^{\text{post}}\text{CBFR}^{\text{Tr}}$ is the time into treatment interaction, and $\varepsilon$ is the random error term.

Data was gathered from both control and treatment groups during two-time periods (2012 and 2018). Subsequently, the net effect was determined using the difference-in-differences (DID) method, the mean gain in the control group from the mean gain in the treatment group. By employing this method, any inherent bias when comparing the control and treatment groups in the CBFR program is accounted for and removed. The DID estimate provided an indication of the impact of the CBFR program on aboveground woody biomass and carbon stock, revealing whether the program had a positive or negative effect.

## RESULTS

### Woody species diversity

We identified 131 species of woody plants (103 trees, 21 shrubs, and seven species of woody lianas) from 38 families (Fig. 3), across nine semi-evergreen mixed forest habitats. In study sites, the five most abundant woody species were *Phyllanthus polyphyllus*, *Grewia damine*, *Bauhinia tomentosa*, *Mallotus philippensis*, and *Glycosmis pentaphylla*. Economically valued woody trees where found as dominant species were *Cassia auriculata* (flower), *Terminalia bellirica* (medicine/fruits), *Manilkara hexandra* (fruits), *Drypetes sepiaria* (fruit), *Phyllanthus emblica* (fruits) and *Pterospertmum suberifolium* (timber). The SMRBs recorded practically all woody species, but the CMRBs recorded only 67. The results indicated that prior to the implementation of the CBFR program, SMRBs had 2 to 64 woody trees per plot with a mean of 24.8, whereas in CMRBs per plot, the number of trees ranged from 1 to 27 trees with a mean of 12.4. After CBFR, a single SMRBs plot had 2

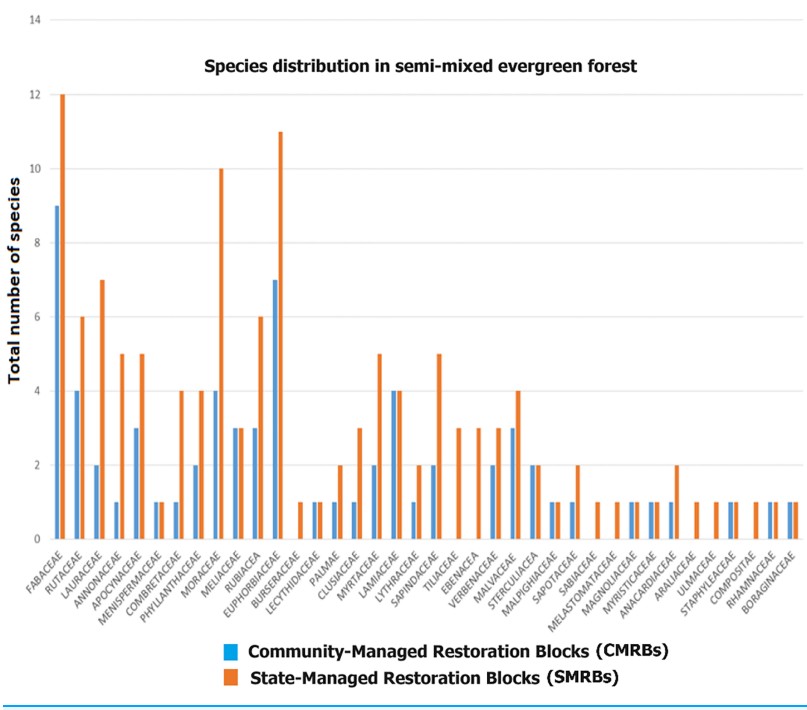

**Figure 3 Species distribution in semi-mixed evergreen forest.**

to 66 trees with a mean of 25.7, and a single CMRBs plot had 2 to 30 trees with a mean of 14.1.

The Shannon diversity index for woody trees showed higher and relatively consistent values in the SMRBs (4.52 in both 2012 and 2018) compared to the CMRBs (3.80 in 2012 and 3.91 in 2018). The DID estimation of woody species diversity provides evidence of the positive impact of the CBFR program on enhancing woody plant diversity; however, it was not significant ($p = 0.12$). Forest management data from Range Forest Offices (Kurunegala, Siyabalanduwa, Mahiyangana, Teldeniya, and Hunnasgiriya), where community forest sites were established, revealed that community members had a preference for species found in CMRBs. Economically valuable plants, such as *Pterospermum suberifolium* for timber, *Michelia champaca, Tectona grandis,* and *Phyllanthus emblica* (Indian gooseberry) for fruit, and *Khaya senegalensis*, emerged as the preferred commercial plantation species among community members because of high local market demand.

## Woody species density and structure

As shown in Fig. 4, the trees within the DBH class of 5–10 cm exhibited the highest density in the community-managed forest, increasing from 621 to 659 per 90 plots between 2012 and 2018. The density of trees within the DBH class of 11–20 cm also increased from 277 to 379, while the DBH class of 21–30 cm increased from 122 to 133, and the DBH class of 31–40 cm increased from 69 to 71. Additionally, the DBH class of 41–50 cm increased from 22 to 24. However, the last DBH class (>50 cm) showed no difference, remaining at 11 throughout the years 2012 to 2018. The basal area of trees in community-managed

forest sites showed variations over the time periods of 2012 and 2018. In 2012, the highest basal area was observed in the 31–40 cm DBH class, whereas in 2018, the highest basal area was recorded in the 11–20 cm DBH class. Conversely, the lowest basal area was recorded in the 5–10 cm DBH class for both time periods (2012–2018).

Similar to the CMRBs, trees of DBH class 5–10 cm which occurred in the state-managed forest blocks, had the highest density and increased from 1,159 to 1,363 per 90 plots from 2012 to 2018. The trees density of DBH class 11–20 cm increased from 549 to 652, while DBH class 21–30 cm increased from 201 to 233, and DBH class 31–40 cm decreased from 155 to 144. Additionally, the DBH class 41–50 cm showed a marginal change from 80 to 82. Finally, DBH class >50 cm had increased from 46 to 51 from the year 2012 to 2018. The basal area of trees was highest in the 31–40 cm DBH class in 2012 and the >50 cm DBH class in 2018, while it was lowest in the DBH class of 5–10 cm in both time periods (2012–2018) for state-managed forest sites (Fig. 5).

Consider the finding of DID estimation of the woody species density among the different DBH classes indicated that the CBFR program had a positive impact on woody species density. However, it was not significant ($p = 0.81$).

### Aboveground woody tree biomass and carbon stock

Table 2 shows the biomass density and carbon density of woody trees in CMRBs and SMRBs in nine forest reserves in the semi-mixed evergreen forest. Tree biomass and carbon density were disproportionally distributed across the nine different forest reserves, with low biomass (18.63 and 21.44 Mg ha$^{-1}$) and carbon density (8.75 and 10.08 Mg C ha$^{-1}$) estimated in CMRBs in Gedaboyaya forest (Fig. 6), and high biomass (98.92 and 106.64 Mg ha$^{-1}$) and carbon density (46.49 and 50.11 Mg C ha$^{-1}$) estimated in SMRBs in Bambarabedda Weliketiya Mukalana forest reserve (Fig. 7).

However, CMRBs in Madigala reserve represent the highest biomass (56.53 and 59.92 Mg ha$^{-1}$) and carbon (26.57 and 28.16 Mg C ha$^{-1}$) density. Moreover, the overall result of biomass and carbon estimates were higher in all SMRBs in the nine different forest reserves than in the CMRBs.

The DID coefficient estimate (Table 3) showed that the CBFR program itself did not have a significant impact on the biomass density and carbon density in the semi-mixed evergreen forest. The average aboveground woody tree biomass for the SMRBs in the semi-mixed evergreen forest was 76.6 Mg ha$^{-1}$ before CBFR and increased to 83.30 Mg ha$^{-1}$ after CBFR. In SMRBs, the average carbon density was 36.02 Mg C ha$^{-1}$ before CBFR and rose to 39.14 Mg C ha$^{-1}$ after CBFR. The CMRBs show significantly less biomass and carbon density in both the before (31.84 Mg ha$^{-1}$ and 14.96 Mg C ha$^{-1}$) and after (35.18 Mg ha$^{-1}$ and 16.53 Mg C ha$^{-1}$) the CBFR program, respectively.

## DISCUSSION

Examining the impact of CBFR on forest species diversity and carbon stocks provides insights into the carbon sequestration potential of the participatory approach with varying floral diversity, and sheds light on the underlying factors considered in further improvement of community-based forest management (Vance-Chalcraft et al., 2010).

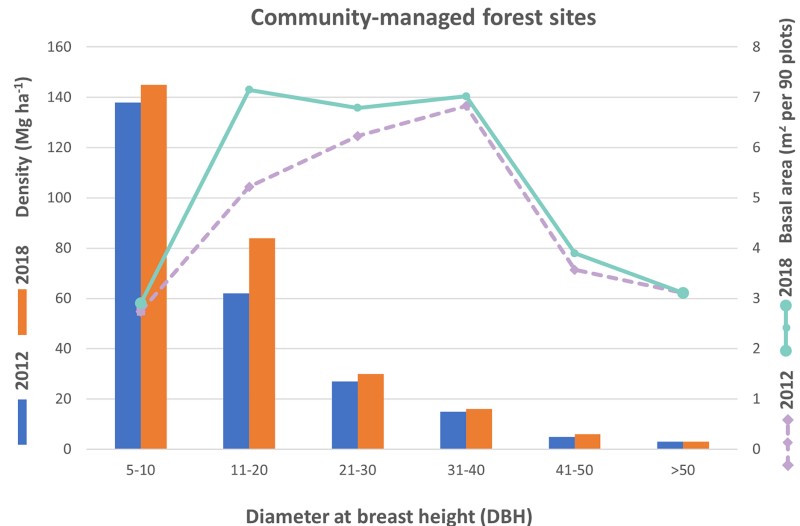

**Figure 4 Trees density and basal area based on diameter distribution in the CMRBs.**

Several tropical forest studies have supported the significance of CBFR in promoting woody plant species diversity, structure, and aboveground carbon stock (*Kumar et al., 2015*; *Mugwedi et al., 2017*; *Gregorio et al., 2020*). In contrast, *Anup (2017)* found that community-managed forests were significantly poorer in species richness and diversity compared to non-community managed forests. This was due to free intense grazing and the special conservation attention given to commercially important trees by forest users in some community-managed forests. This study result also provides the valuable funding to broaden the scope of community based forestry and fill the data gap in the semi-mixed evergreen forest.

Our results revealed greater woody species representation in SMRBs compared to CMRBs in year 2012 as well as 2018 but the species diversity did not change within the 5 years period. However, the DID estimate indicated that the CBFR program increased woody species diversity in the CMRBs. The reason for this is most of the CMRBs are degraded forest land with few numbers of scattered trees. Therefore, at the initial stage, the species diversity was considerably low. After implementing the CBFR program, community members planting different tree species in the CMRBs, and woody species diversity increased over time compared to the SMRBs. The diversity, distribution, and population structure of tree species provide fundamental information for forest conservation and management (*Sahu, Dhal & Mohanty, 2012*; *Farooq et al., 2019b*). Lower values of species diversity indicate that one or a small number of species dominate a given area (*Ifo et al., 2016*; *Farooq et al., 2022a*, *2022b*). During the 5-year sampling period of our study, the impact of CBFR on woody species diversity was found to be insignificant, which is similar to findings from a previous study conducted in Cambodia, where control sites exhibited a greater abundance of tree species, whereas CBFR sites were implemented two or 5 years prior to sampling (*Lambrick et al., 2014*). Conversely, research conducted in Ethiopia found that community-managed forest exhibited higher woody species diversity

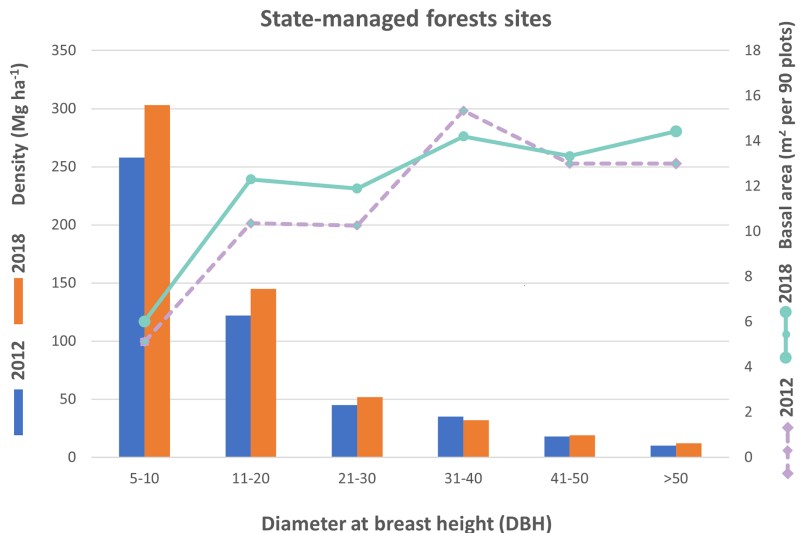

**Figure 5 Trees density and basal area based on diameter distribution in the SMRBs.**

**Table 2 Total biomass density and carbon density of woody trees in nine forest reserves.**

| Name of forests | Biomass density (Mg ha$^{-1}$) (Carbon density (Mg C ha$^{-1}$)) | | | |
|---|---|---|---|---|
| | Before/CMRBs | After/CMRBs | Before/SMRBs | After SMRBs |
| Bambarabedda Weliketiya Mukalana | 27.10 (12.73) | 29.08 (13.66) | 98.92 (46.49) | 106.64 (50.11) |
| Galkanda | 33.76 (15.86) | 38.50 (18.09) | 72.85 (34.24) | 80.31 (37.74) |
| Hawanarawa | 20.73 (9.74) | 23.41 (11.00) | 81.38 (38.24) | 89.46 (42.04) |
| Rakaula | 43.24 (20.33) | 48.38 (22.74) | 70.76 (33.26) | 76.87 (36.13) |
| Dolukanda | 21.48 (10.09) | 24.61 (11.56) | 71.86 (33.77) | 79.76 (37.48) |
| Dunukewala | 36.11 (16.97) | 39.10 (18.38) | 71.88 (33.78) | 77.03 (36.20) |
| Gedaboyaya | 18.63 (8.75) | 21.44 (10.08) | 65.60 (30.83) | 69.32 (32.58) |
| Walasgala | 29.01 (13.63) | 32.18 (15.12) | 68.86 (32.36) | 76.58 (35.99) |
| Madigala | 56.53 (26.57) | 59.92 (28.16) | 87.79 (41.26) | 93.77 (44.07) |

**Note:**
Carbon density (Mg C ha$^{-1}$) are reported in parentheses.

and evenness compared to non-community managed forest. This difference is likely due to human disturbances, such as illegal extraction of forest resources by local people and livestock grazing, which can lead to the dominance of certain species in non-community managed forest (*Tiki et al., 2024*). Our study observation and records of Department of forest conservation highlighted that t CMRBs are typically situated in greater proximity to village areas in comparison to SMRBs (*Ekanayake, Cirella & Xie, 2020*). Being near residential areas has a greater influence on the tree composition of CMRBs as people depend on those forest patches for their needs (*Ekanayake, Cirella & Xie, 2020*). It is worth noting that several studies have shown that the distance between the forest and a nearby settlement can have a significant impact on forest composition, including density and diversity. For instance, a recent study conducted by *Hending et al. (2023)* found that

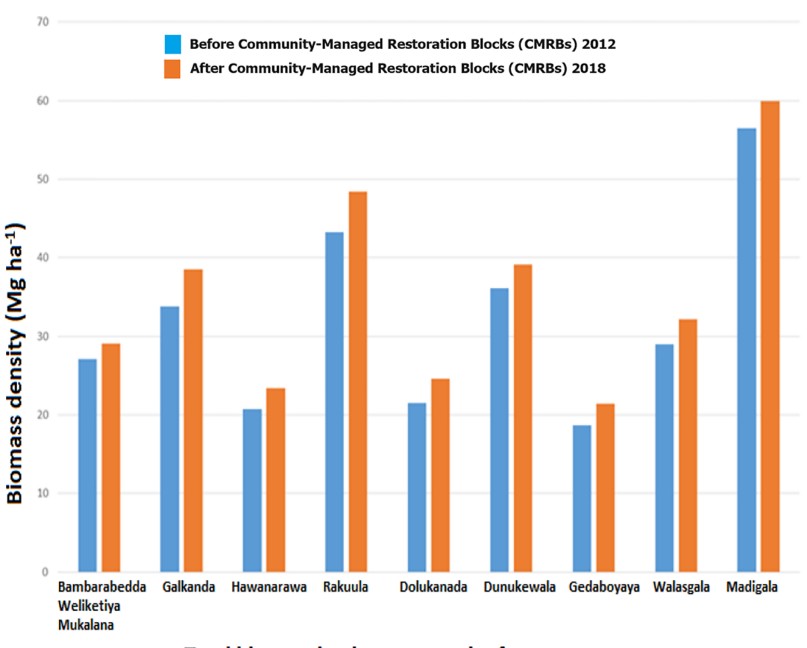

**Figure 6** Total biomass in the community-managed forest reserves.

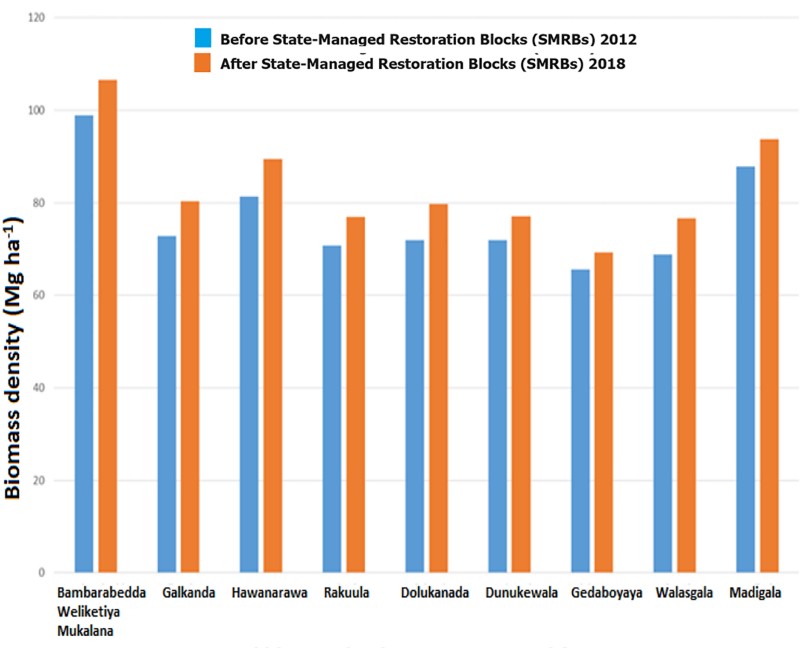

**Figure 7** Total biomass in the state-managed forest reserves.

**Table 3 The DID estimates of biomass density and carbon density of woody trees in the semi-mixed evergreen forest in Sri Lanka.**

| Variable | CMRBs | | SMRBs | | DID estimation result | |
|---|---|---|---|---|---|---|
| | Before the CBFR program | After the CBFR program | Before the CBFR program | After the CBFR program | Coefficient | $p > (t)$ |
| Biomass density (Mg ha$^{-1}$) | 31.84 | 35.18 | 76.6 | 83.30 | −3.32 | 0.675 |
| Carbon density (Mg C ha$^{-1}$) | 14.96 | 16.53 | 36.02 | 39.14 | −1.55 | 0.676 |

species diversity, tree size, and structural diversity were significantly reduced in the forest edges of Madagascar's transitional forests which are mainly influenced by human activities.

Tree density and basal area are vital woody species composition information that describe the structure of a forest (*Tarin et al., 2017*; *Farooq et al., 2019a*, *2020*). The analysis of diameter distributions showed smaller tree size and basal area in the CMRBs blocks than in the SMBRs which may be due to the younger age of the planted trees. The higher tree density and young age might affect the basal area of CMRBs, as site index and density are related to the basal area. In general, site index, crown size, stand age, and stand density influence the basal area of tree species (*Deetlefs, 1954*). Similar to our study, a study conducted by *Paudyal et al. (2017)* in Nepal revealed that the number of pole-sized trees (small trees) increased in community managed sites while that of mature trees decreased due to harvesting for commercial purpose. Thus, overall basal area is less in those Community managed forest. In contrast, co-managed mangrove forests in Kenya demonstrated significantly higher basal areas and standing densities compared to state-managed forests (*Kairu et al., 2021*).

Several studies conducted worldwide revealed that the implications of CBFR programs have brought about positive changes in landscape restoration (*Gregorio et al., 2020*) and forest management (*Gurung et al., 2013*). A recent study by *Luintel, Bluffstone & Scheller (2018)* found that Nepal's community forestry had a significant positive impact on increasing the forest cover of degraded natural forests and biodiversity conservation compared to non-community forestry sites. In addition, *Pandit & Bevilacqua (2011)* and *Bijaya et al. (2016)* explored how CBFR improved forest tree density by planting economically and ecologically valuable trees at community management sites. Some studies have found that community forestry programs exhibit the capacity to improve and conserve the forest ecological system by increasing the structure and composition of the existing landscape (*Yadav et al., 2003*; *Chinangwa, Pullin & Hockley, 2017*; *De Jong et al., 2018*; *Ojha et al., 2022*).

CBFR sites located in Badulla (Gedaboyaya) and Monaaragala (Hawannarawa) districts are predominantly finger millet (*Eleusine coracana*) producing areas. Finger millet cultivations were observed as shifting cultivations (Chena) in natural forests since late 1980 (*Marambe et al., 2020*). As this crop requires more sunlight and is resistant to drought, people used to clear the forest land by removing larger shade trees. Usually, these Chena

lands are located at the edge of the forest and are converted into CMRBs. Therefore, CMRBs in Gedaboyaya forest reported the lowest biomass and carbon density compared to the other forest.

The high biomass and high carbon density are attributed both to the large number of individuals and the presence of bigger trees (*Wang et al., 2023*). Large trees (high DBH) were scarce in the CMRBs due to removal through Chena cultivation and harvesting for wood for fuel, though they stock most of the estimated biomass. Conversely, low-biomass trees (less DBH) dominated in CMRBs. The primary components of total biomass are the live tree and the dead wood component. The current study only focused on the woody tree biomass due to a shortage of data related to variables such as dead wood factors (dead trees, down litter). Therefore, the estimated result of our study on the carbon density of natural forests (36.02 Mg C ha$^{-1}$ before CBFR and 39.14 Mg C ha$^{-1}$ after CBFR) was relatively limited when compared to studies that reported the total aboveground carbon density of natural forests in various tropical countries, *e.g.*, in the Philippines (86 Mg C ha$^{-1}$) Malaysia (100 Mg C ha$^{-1}$) and Thailand (98.76 Mg C ha$^{-1}$) (*Pibumrung, Gajaseni & Popan, 2008*). Moreover, a study conducted in Sinharaja forest of Sri Lanka revealed that the total aboveground carbon content per hectare is 237.2 metric tons (*Nissanka & Pathinayake, 2009*). However, a study conducted in Tankawati, a natural hill forest located in a moist tropical climate region of Bangladesh, reported a higher carbon density of woody tree biomass (110.94 Mg C ha$^{-1}$) than the findings in the current study (*Ullah & Al-Amin, 2012*). These values are higher as a result of the greater richness of vegetation composition and a denser canopy structure than in semi-mixed evergreen forest. However, a study conducted in the Udawattakele Forest Reserve (UFR) in the Kandy District of Sri Lanka found that the total carbon density of the tree biomass was 36.55 Mg C ha$^{-1}$. This result is quite similar to our findings.

The DID coefficient estimation in this study revealed that the CBFR program did not have a significant influence on the carbon stock of aboveground woody tree biomass. Consistent with our findings, a study by *Luintel, Bluffstone & Scheller (2018)* also demonstrates a notable negative impact of community forestry on the aboveground carbon (AGC) of trees and saplings at a national level. This study finding contrasts with other studies that have reported significant higher carbon density in community-managed forest than was the case with state managed forest (*Gurung et al., 2015*; *Basnet et al., 2018*). However, the effect of community-based forestry on carbon stocks varies across different geographic and topographic contexts, as well as in forests with different canopy structures. Specifically, no significant effects of community-based forest on aboveground carbon were observed at lower elevations, in the Terai or hill regions, or beneath dense canopies (*Gurung et al., 2015*). As mentioned above, a short sampling period (5 years) is not enough to maintain a higher tree density (high biomass) in CMRBs. Also, in community-managed woodlots, farmers practice thinning operations to reduce crowding and competition among the trees and to maintain a steady growth rate. Therefore, tree density was less in the CMRBs. Similar to our findings, a study conducted in six different community forests in the Dolakha district, Nepal, found that, due to the exclusion of low-diameter trees, the community forest sites had lower biomass and carbon densities (*Shrestha et al., 2013*).

Moreover, a study conducted in tropical dry deciduous forest ecosystems in northwestern Himalaya reported that reserve forest or the forest completely under control of state has maximum carbon density (69.15 Mg C ha−1), whereas the minimum was recorded in a co-operative society forest (CSF) (33.27 Mg C ha$^{-1}$) or the forest managed by the community (*Kumar et al., 2022*).

## CONCLUSIONS

The present study enhances the understanding of Community-Based Forest Restoration by evaluating the effects of the CBFR program in a semi-mixed evergreen forest in Sri Lanka, a forest type with limited national and global research. Our research demonstrates that CBFR impacts tree diversity, density, total biomass, and carbon stock, depending on the pre-existing conditions of the forest. Tree diversity and density were significantly higher in SMRBs compared to CMRBs due to the robust protective rules and regulations implemented by the Department of Forest Conservation. Additionally, our results indicated that the average aboveground woody tree biomass in SMRBs was twice as high as in CMRBs. CMRBs exhibited lower biomass and carbon density both before and after the CBFR program. Therefore, our findings revealed that the CBFR program does not significantly influence carbon density and woody tree biomass due to its short implementation period. However, this study did not collect tree height data as a parameter, preventing the use of improved allometric models to estimate aboveground biomass. To minimize bias, the development of locally derived diameter–height relationships is suggested. Moreover, this study did not assess the impact of changes in below-ground biomass and soil organic carbon. Therefore, broadening the scope with additional empirical research on both above-ground and below-ground biomass at different periods, locations, and scales is needed.

## ACKNOWLEDGEMENTS

The authors gratefully acknowledge Ms. Robyn P. Geldard for English proofreading, and all officers in the Department of Forest Conservation in Sri Lanka for their support during data collection.

### Funding

This work was supported by the National Natural Science Foundation of China (No. 72063006), the National Social Science Foundation of China (No. 21AZD058), and the Natural Science Foundation of Hainan Province (No. 720MS027). The funders had no role in study design, data collection and analysis, decision to publish, or preparation of the manuscript.

### Grant Disclosures

The following grant information was disclosed by the authors:
National Natural Science Foundation of China: 72063006.

National Social Science Foundation of China: 21AZD058.
Natural Science Foundation of Hainan Province: 720MS027.

## Competing Interests

The authors declare that they have no competing interests.

## Author Contributions

- Shahzad Ahmad conceived and designed the experiments, performed the experiments, analyzed the data, prepared figures and/or tables, and approved the final draft.
- Haiping Xu conceived and designed the experiments, performed the experiments, authored or reviewed drafts of the article, and approved the final draft.
- E. M. B. P. Ekanayake conceived and designed the experiments, performed the experiments, analyzed the data, authored or reviewed drafts of the article, and approved the final draft.

## Data Availability

The raw data is available in the Supplemental File.

## Supplemental Information

Supplemental information for this article can be found online at http://dx.doi.org/10.7717/peerj.18176#supplemental-information.

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
