# Peer review of "Impact of community-based forest restoration on stand structural attributes, aboveground biomass and carbon stock compared to state-managed forests in tropical ecosystems of Sri Lanka"

_PeerJ, doi:10.7717/peerj.18176_

## Round 0.1 · original submission · Major Revisions

Reviewer 1 suggests that there are better methods that you should have used, please either make these changes or explain clearly why their concerns are not well-founded. Reviewer 2 also provides comments and suggestions that could improve the manuscript if answered carefully.

Reviewer 1 ·

Basic reporting

1) The English is clear, and the text is professional and well-written. I only suggest that some information be summarized.
For example,
lines 30-37 in the abstract: I recommend that the authors provide only the average between the two types of treatment. It is not necessary to include so many details in an abstract.

2) The authors describe the objectives in lines 146-151; however, a hypothesis or research question is missing to clarify the purpose of the study. Especially since the title begins with the word "impact," there needs to be an explanation of why this impact is being measured. It should go beyond a merely descriptive study, as currently presented.

3) The authors also repeatedly describe the acronyms in several instances after initially defining them, for example, in lines 211-212. I suggest doing this only once in the text.

Experimental design

Here are my main concerns regarding the manuscript.

Lines 249-252 - I suggest deleting this section. I recommend that the authors simply report the statistics and justify the use of the test, as I do not see the need for these detailed descriptions.

Line 266 - The use of the allometric equation recommended by Brown et al. (1989) is not incorrect, but it is not the most recommended approach for estimating aboveground biomass stocks in tropical forests today. This equation does not include species-specific wood density or estimated height of tree individuals. It was developed in the 1980s. Current models such as the pantropical allometric equation by Chave et al. (2014) or the wood density database by Zanne et al. (2019) provide more accurate estimations with updated data on wood densities.

In addition to those, the Pearson protocol, Walker and Brown (2005) , presents more current models. Interestingly, one of the authors is the same author of the equation used by the authors here.
Pearson T, Walker S, Brown S (2005) Source book for land use, land-use change
and forestry projects. Winrock International, Arlington, TX

Lines 297-308 - I suggest resolving this with a repeated measures ANOVA, wouldn't that make more sense? There are several modern models that calculate these differences as factors. Tests like Dunnett's can set control areas against treatment areas.

Validity of the findings

I suggest replacing the equation and using more robust statistical analyses, as I mentioned in the previous topic, to validate the conclusions presented here. For that, the authors need to define the article's focus. There is a blend of social impact and ecological field analyses, and there isn't a clear focus profile.

Additional comments

Figures

The figures are good, but all captions are missing the following: study area, sampling units, definition of acronyms. Figures should have captions that make them understandable without needing to refer back to the text.

·

Basic reporting

The literature is reviewed in the introduction, but the location of the studies is not mentioned, so we can't say whether they are relevant to this research problem.

The authors need to explore more literature to contrast the findings in the discussion section.

The citations should be arranged according to journal format in chronological order, and appropriate parentheses should be used when mentioning the citations at the beginning or middle of the sentences.

The language is good enough for readers to understand, except for some grammatical corrections like missing commas.

Experimental design

In the data collection section, author failed to describe what parameters were measured using what instruments.

It is mentioned the data collected between 2012 and 2018 but the date in supplementary data is 2013.

Validity of the findings

Some of the justifications are not supported by objectives, such as how the distance from villages affected the diversity.

Limitations are not mentioned.

The conclusion should be re-written and shorten including only important findings.

Additional comments

The research provided important information regarding the Community-based restoration in the tropical region of Sri Lanka. Such studies should be replicated on a larger or global scale before reaching the exact conclusion of such a study.

I have provided my comments on the "peerj-reviewing-99692-v0_reviewed" file.

Also here:
Comments
Abstract
Line 21: was the initial data of 2012 or 2013? I found the date 2013 in data sheet.
Line 22: add compared to state-managed....
Line 37: carbon density together
Line 45: Use the words not present in the title. Split “ Woody species diversity and density” into two
Introduction:
It was well-written and provided sufficient literature.
Line 50: only rural? what about other regions? merge first and second sentence and re-phrase.
Line 54: You jumped from one senario to tropical.
Line 58: what sorts of studies? provide some references.
Line 66: use alternatives of moreover.
Line 67: any references?
Line 70: merge these sentences and re-phrase.
Line 74: Define CBFR. Use the acronym only after writing the full form.
Line 80-99: In which region of the world these studies were conducted? Are these from tropical regions? Are the cases similar to your region?
Line 115: are
Line 122: arrange the citations chronologically.
Line 124: arrange the citations chronologically.
Line 137: Define woody species.
Line 139: why is studying species diversity, density, biomass, carbon stock important? Is this due to no previous study? What's the significance?
Line 142: Is it?

Materials and Methods
Study area:
Line 164: use the consitant unit.
Line 165: add comma.
Line 166: you already mentioned the intermediate zone; instead, write "study area".
Line 167: add and.
Line 168: use alternative; don't repeat. What about the area important for wildlife species?
Line 170: Citation?
Line 174: forest?
Line 178: What's the population of the area?
Line 180-184: merge these sentences to the sentence where you mentioned your study area.
Figure 1: I think it is better to show the climatic zone map of Sri-Lanka. Provide the source from where you downloaded the boundary. \
Line 192: How much?
Line 192-195: repeated information.
Line 203: Use sentence case. Do not capitalize each word.
Data collection:
Line 208: any other studies that used a similar method?
Line 213: full stop. was the initial data of 2012 or 2013? I found the date 2013 in data sheet.
Line 218: evenly? you mentioned depends on the stand's uniformity and size.
Line 220-223: Mention what parameters of the tree you measured. what instruments did you use?
Data analysis:
Line 241: multiple full stops?
Line 249: Shannon-Weiner? It is H’
Line 307: do not write the word like as following as below

Results:
Line 334-337: I wonder how you mentioned the exact or average number of tree count. You didn't mention the counting of trees in the method section. You should clearly mention what you recorded from the field and how.
Line 347: mention the p-values.
Line 356-358: Did you mention the DBH class division for trees in your data collection or research design section?
Line 362: this interpretation should be written in discussion.
Line 387: mention the p-values.
Line 399: 2 or 3?
Line 412: before/ after CBFR?

Discussion:
Line 438: You mentioned the species preferences of the people for plantation. Doesn't this reduce species diversity?
Line 447: was it from a tropical region?
Line 450: Did you mention it in the result?
Line 452: Find more literature and contrast. or is this relevant to your findings or your objectives? did you measure the distances that affect the diversity?
Line 462: Find more literature and contrast.
Line 496: change the unit
Line 508: citation?
Line 523: mention the limitations of your study in the last separate paragraph

Conclusion:
This section should re-state the important findings only based on your research and provide relevant future recommendations.
Line 526: Remove the first sentence.
Line 531: what does it mean?
Line 532: is it your finding? or discussion?
Line 538: don’t mention the data.

References:
Double-check that all the citations are present in the references.
Provide DOIs if available
Check the reference style of PeerJ.

---

## Round 0.2 · accepted · Accept

Thank you for your careful replies to all of the comments. I am satisfied that you have responded adequately based on my reading of the manuscript. The manuscript can now be accepted for publication.